# Future Academic Expectations and Their Relationship with Motivation, Satisfaction of Psychological Needs, Responsibility, and School Social Climate: Gender and Educational Stage

**DOI:** 10.3390/ijerph18094558

**Published:** 2021-04-25

**Authors:** David Manzano-Sánchez, Alberto Gómez-Mármol, Luis Conte Marín, José Francisco Jiménez-Parra, Alfonso Valero-Valenzuela

**Affiliations:** 1Department of Physical Activity and Sport, Faculty of Sport Sciences, University of Murcia, Santiago de la Ribera, 30720 Murcia, Spain; david.manzano@um.es (D.M.-S.); conte@um.es (L.C.M.); avalero@um.es (A.V.-V.); 2Department of Didactics of Plastic, Musical and Dynamic Expression, Faculty of Education, University of Murcia, Espinardo, 30100 Murcia, Spain; alberto.gomez1@um.es

**Keywords:** secondary education, elementary education, teenagers, children

## Abstract

The purpose of this study was to show the relationship between future academic expectations and the concepts of motivation, basic psychological needs, school social climate, and responsibility. Additionally, possible differences in future expectations were analyzed according to gender and educational stage. A total of 984 students (M = 12.87; SD = 1.84) from elementary and secondary school participated in this study. A single questionnaire composed of different scales was administered to check the values of motivation (EME), psychological need satisfaction (PNSE), school social climate (CECSCE), responsibility (PSRQ), sociodemographic differences, and a question to find out future academic expectations. Results showed that the group with the highest future expectations (do a degree or upper vocational training) had statistical differences of *p <* 0.001 with respect to the group with the lowest future expectations (finish compulsory secondary studies or basic vocational training and start work) and middle expectations (do a high school or middle vocational training) with regard to autonomous motivation, satisfaction of psychological needs, school and teacher climate, and social and personal responsibility. The group with the lowest expectations had higher values with respect to the other two groups in terms of amotivation (*p <* 0.001). Moreover, women and elementary school children had greater future academic expectations (*p <* 0.001). In conclusion, the promotion of basic psychological needs satisfaction, autonomous motivation, personal and social responsibility, and school social climate are related to higher academic expectations, and the improvement of these variables becomes especially important for boys and secondary students who could have a higher chance of dropping out of school.

## 1. Introduction

The current Spanish educational system is mainly oriented towards providing students with tools to achieve comprehensive development and significant learning through the acquisition of skills that facilitate their adaptation to society [1], and improve their academic, social, cognitive, and emotional performance. This is due to the great social demand that has existed for years, regarding the need for standard education to provide schoolchildren with sufficient tools to be able to adapt to society’s constant changes [2].

One of the main concerns of international education policies is the so-called ‘early school leaving’, defined by the European Commission (EC) as the moment when a person reaches the first cycle of secondary education and decides not to continue with his/her academic training [3], which can have a direct impact on the personal and social development of students. Moreover, this aspect becomes one of the main indicators of social exclusion [4]. In this sense, the Europe 2030 strategy projected by the EC has the objective of reducing early school leaving to a maximum of 10% in each of the countries of the European Union (EU) [5]. However, Spain is one of the countries that did not yet meet this threshold, with a percentage of 17.9% [6]. 

Studies in this line of research determine that school dropout is a multi-causal phenomenon [7] that involves several relevant risk factors [6,8] related to young people (e.g., students’ academic and professional expectations or school failure), to the family (e.g., demographic factors), to the community (e.g., geographical location), and to the school (e.g., resources or program diversity). Other research that investigated the variables that influence school dropout [9] found that gender, socio-economic background, family, and beliefs play a key role. In turn, others [10,11] consider individual factors (e.g., attitudes, self-regulation, self-perception, life goals, school involvement, and future expectations) are the main risk elements.

In this line, school failure, understood as the difficulties students have in meeting educational objectives [12], is considered one of the main factors associated with and predicting school drop-out [8,13]. On the other hand, students’ future academic expectations, defined as a person’s beliefs that a particular event will actually occur in the future, have a strong influence on target setting [14] and show a negative correlation with dropout [11]. The positive association between future expectations and self-perception allows a person who imagines a more positive future to have better academic results [15]. 

Within the educational context, one of the most widely used theoretical frameworks for understanding students’ motivational processes is the Self-Determination Theory (SDT) [16,17]. This macro-theory focuses on the functioning of human motivation in different social contexts, constituting a continuum that goes from more autonomous to more controlling motivational states [18] and that influence the psychological well-being of students [17]. In the framework of SDT, Vallerand’s [19] hierarchical model indicates that the impact of social factors (e.g., responsibility or school social climate) is mediated by basic psychological needs (autonomy, competence, and relatedness) that are essential and inherent to each person [20], whose satisfaction or frustration of these needs can lead to a more or less self-determined motivation and generate different cognitive, behavioral, and affective consequences in people’s lives [18]. In this line, research related to the assignment of autonomy and responsibility to students [21], found a key factor that can satisfy basic psychological needs [22], the school social climate [23], and experience of fun in class [24].

Responsibility, understood as the ability to face the consequences of one’s own decisions [25], shows a positive correlation with self-determined motivation and the satisfaction of basic psychological needs [26,27]. Other studies [27,28] found that responsibility is a capacity that can predict these variables.

In this sense, Manzano-Sánchez and Valero-Valenzuela [2] suggest the need to establish a direct relationship between responsibility and the school social climate. In recent years, the number of studies that analyzed this variable increased [23,29], finding a direct relationship with academic performance, intrinsic motivation, and basic psychological needs. Various research studies [11,30] indicate the importance of effective classroom management by teachers in creating a good learning climate and reducing both drop-out and failure. Therefore, the classroom climate can be considered a key element for increased academic and social performance [31].

Although there are many studies that analyzed future expectations in the context of other self-perceptions [32], no study so far has linked this variable to motivational processes, basic psychological needs, responsibility, and the school social climate. Other studies that analyzed the relationship of other self-perceptions with responsibility, school social climate, motivation, and basic psychological needs in the educational context [2,21,33], found the effectiveness of these variables to improve students’ self-concept, self-efficacy, and self-esteem [34]. Given the direct relationship of self-perception and self-esteem with future academic expectations [32], it could be considered that an improvement in psychosocial variables (responsibility, classroom climate, motivation, and basic psychological needs) would also generate greater self-perception and better future expectations in students. However, this is an aspect that still needs to be addressed and investigated in depth from the scientific point of view.

Depending on the theoretical basis presented, the main objective of this study was to carry out a profile analysis generating groupings according to future expectations, in order to assess how they relate to motivation, basic psychological needs, responsibility, and the school social climate. In addition, a secondary objective was set, based on checking the possible differences in future expectations according to the gender of students and their educational stage.

## 2. Materials and Methods

### 2.1. Study Design and Participants

A transversal and quantitative study was carried out and informed consents were requested from the students and their parents. The ethical criteria were fulfilled by obtaining a favorable report from the Ethics Committee of the University of Murcia (registration number 1685), which allowed this research to be carried out.

A total of 984 students from two Elementary Schools and two Secondary Schools participated in this study. The sample was chosen through accessibility and convenience. These were from centers with similar socio-demographic characteristics and they were between nine and 16 years old (M = 12.87, SD = 1.84). More specifically, there were 246 students from elementary school (114 men and 132 women) and 738 from secondary school (417 men and 321 women). The inclusion criteria used to participate in the research were the following—similar socio-demographic characteristics and completion of all questionnaires. On the other hand, a statistical test was carried out to delete possible atypical answers and inadequate response patterns.

### 2.2. Procedure

First, a bibliographical review was carried out and contact was made with the main research centers. After that, the participants were given an information sheet and were asked to sign an informed consent form. Subsequently, they were informed of the purpose of the research and were told that participation was voluntary and confidential. Students attended an oral presentation on how to complete the questionnaires with an explanation provided by one of the researchers. After that, the teacher read the items of each scale in order to ensure their understanding. The questionnaires were administered in the specific classroom of each grade within their school, in a calm environment, and without the presence of possible distractions. The teacher and one of the researchers stayed with them all the time solving possible doubt. The participants were requested to provide true answers. The test was completed by each class-group in approximately 30 min; all students filled out the form at the same time.

### 2.3. Instruments

A closed-question questionnaire was used in the present study, it had two parts, the first one consisted of socio-demographic variables and the second part were the scales used in the study.

Socio-demographic variables included gender, birthdate, and a closed question, in order to know what the academic future expectations of the students were. This question had three possible answers—(a) finish secondary compulsory studies (SCS) or basic vocational training (BVF) and start to work; (b) complete high school or middle vocational training (MVF); and (c) complete a degree or superior vocational training (SVF).

(1) Personal and Social Responsibility Questionnaire (PSRQ)—to measure personal and social responsibility levels. It was adapted to the school context by Li et al. [35] and for Spanish by Escartí et al. [36] and validated in a 9- to 15-year old sample similar to our sample. This scale consisted of 14 items, seven to assess social responsibility and seven for personal responsibility. The answers were provided on a Likert-type scale ranging from 1 (totally disagree) to 6 (totally agree). Reliability in the pre-test was 0.844 for social responsibility and 0.744 for personal responsibility.

(2) Psychological Need Satisfaction in Exercise (PNSE)—to measure the satisfaction of the needs of social competence, autonomy, and relatedness. The scale was adapted for Spanish and to the education context by Moreno et al. [37], and was validated in a 12–16 year old sample. This scale consisted of 18 items, six to evaluate each need—competence, autonomy, and relationships with others. These were preceded by the sentence “During my class…” and the answers were provided on a Likert-type scale ranging from 1 (False) to 6 (True). Reliability in the pre-test was 0.565 for autonomy, 0.694 for competence, and 0.724 for relationships. Moreover, the psychological mediator index (PMI) was applied to jointly evaluate the three variables, yielding an internal consistency of 0.775.

(3) Motivation toward Education Scale (in French, EME)—to measure motivation from the most self-determined types to the most external causes and amotivation. The Spanish version of the Échelle de Motivation en Éducation [38] validated by Nuñez et al. [39] was used. The questionnaire passed a reliability test in order to check the understanding of the student sample in the same way as the others. The questionnaire consisted of five subscales, called intrinsic motivation, identified regulation, introjected regulation, external regulation, and amotivation. The instrument was composed of 28 items that preceded by the sentence “I go to school/high school because…”, with a seven-point Likert-type scale, from 1 (totally disagree) to 7 (totally agree). The reliability was 0.905 (intrinsic motivation), 0.644 (identified regulation), 0.735 (introjected regulation), 0.732 (external regulation), and 0.786 (amotivation). These variables were unified in the named “self-determination index” (SDI).

(4) Questionnaire to assess social school climate (CECSCE)—to evaluate the climate perceived by the students with regard to their class, teacher, and school. It was designed by Trianes et al. [40]. The questionnaire consisted of two subscales called “School climate”, made up of eight items, and “Teaching climate” composed of six items. A five-point Likert-type scale was used, ranging from 1 (totally disagree) to 5 (totally agree). The internal consistency analysis yielded a value of 0.851 for school climate and 0.753 for teaching climate.

### 2.4. Statistical Analysis

First, the answer of the participants were recodified in Excel for the socio-demographic questions and after that, all values were copied onto the statistical software IBM SPSS (23.0). After this descriptive statistics (i.e., mean, SD, skewness, and kurtosis) were calculated for the variables considered. Skewness and kurtosis were used to check the normality of data, considering values < 3 and < 10, respectively, as normal values. Cronbach’s alpha coefficient was estimated to examine the reliability of each variable analyzed, which is acceptable with values over 0.70 [41] and for scales with few items, a measurement of 0.50 was enough. [42]. In the second step, we checked the correlation between the variables and these supported the study of Hair et al. [43], with values < 0.80 supporting the absence of multicollinearity between the variables.

In addition to check whether there were significant differences in future academic expectations, a multivariate analysis of variance (MANOVA) was performed, calculating the main effect. Before carrying out the MANOVA, Levene’s test of homogeneity of variance tests was run to check some of the parametric statistical assumptions. Levene’s results were *p <* 0.05 for the most part of the variables (except for school climate, identified, introjected, and external regulation); therefore, we decided to perform the analyses through non-parametric tests as well. The results obtained with both procedures were very similar and the results of the non-parametric tests were not included for brevity. 

In those cases in which a statistically significant difference was found, a post-hoc contrast of comparisons test was affected using the Bonferroni correction, to determine in which clusters there would be statistically significant differences. In addition, the effect size was calculated in terms of partial eta squared (η^2^), considering a small effect size with values = 0.01, medium with values = 0.06, and large with value = 0.14 [44].

Finally, we checked the differences between academic future expectations and gender and if they were for elementary or secondary school, using contingence tables with the chi-square test and residual analysis.

## 3. Results

### 3.1. Descriptive Results

Table 1 presents the mean values, standard deviation, asymmetry, and kurtosis values, reliability of the variables, and the correlations. In this sense, the values of the different scales were correlated with each other. It is important to note that the values above 0.80 were only in the variables that make up the PMI and SDI index, and between the teacher and school climate, which are in the same questionnaire. Amotivation was the only variable that was negatively correlated with all variables, except with autonomy, with which there was no significant correlation. The asymmetry and kurtosis values presented a good value following the recommendations mentioned in the statistical analysis section.

### 3.2. Differences According to Student Future Academic Expectations

The multivariate analysis of variance test (Table 2) showed a statistically significant multivariate effect for the different future academic expectations, Wilk’s Lamda: 0.878; F = 5.433; *p =* 0.000. All variables indicated statistically significant differences in *p <* 0.001, except in introjected regulation, external regulation, and autonomy.

Post-hoc analysis reported the following differences. In intrinsic motivation, (F = 14.310, η^2^ = 0.028), identified regulation (F = 13.743, η^2^ = 0.027), competence (F = 16.353, η^2^ = 0.033), relatedness (F = 9.303, η^2^ = 0.019), school climate (F = 12.397, η^2^ = 0.025), teacher climate (F = 15.007, η^2^ = 0.030), social responsibility (F = 28.239, η^2^ = 0.054), personal responsibility (F = 28.239, η^2^ = 0.054), and PMI (F = 10.453, η^2^ = 0.021), in favour of “Do a degree or SVF” group, and without differences between the other two groups.

On the other hand, there were differences among the three groups in amotivation (F = 25.776, η^2^ = 0.050) with higher values in the group “Finish SCS or BVF and start to work”, middle values in the second group “Do high school or MVF”, and low values in the third group “Do a degree or SVF”. These differences were found in SDI (F = 34.643, η^2^ = 0.066) in favor of “Do a degree or SVF group”, with the middle values in “Do high school or MVF” and low values in the first group “Finish SCS or BVF and start to work”.

### 3.3. Differences According to Gender and Age

Finally, in order to check the differences in the three different groups in terms of gender and the school course, a difference analysis through Pearson’s chi-square statistic was performed. We checked the difference with this test and the use of the corrected typified residuals, since residuals equal to or greater than 1.90 were considered as indicators that there was dependence between these 2 categories and that, therefore, the differences were significant.

In terms of gender, Table 3 reports that there were more boys in the first group “Finish SCS or BVF and start to work” and more women in the third group “Do a degree or SVF”. According to their course level, more elementary school students were in the third group “Do a degree or SVF”, and more secondary students in the other two groups.

## 4. Discussion

This research aimed to analyze future expectations with regard to further study or starting to work, in a sample of elementary and secondary school students. The objective consisted of determining the relationship among these future expectations with motivation, basic psychological needs, personal and social responsibility, and school social climate. In addition, the potential differences according to gender and educational stages were studied.

In this sense, the data analysis reported a positive correlation among autonomous motivation (intrinsic motivation and identified regulation), higher basic psychological needs satisfaction, higher personal and social responsibility development, and a better school climate perception. This correlation between motivation and basic psychological needs satisfaction was already verified by Leo et al. [45] and Kaiser et al. [46], who also found a positive correlation, as this study stated. As Leo et al. [45] remarked, specifically in the educational context, in Physical Education lessons, the teaching methodology used by teachers, conditions whether this correlation exists or not. In turn, the relation between personal and social responsibility and motivation (also from an SDT perspective), was studied by Manzano-Sánchez et al. [26] and Manzano-Sánchez and Valero-Valenzuela [2]. With this regard, both studies concluded as a pedagogical implication that responsibility development (for instance through teaching methodologies such as the Personal and Social Responsibility Model by Hellison [47]) might be a good pedagogical choice, if increasing motivation in the school context was targeted.

In the same manner, in school climate analysis, several pieces of research affirmed that it was mediated by other variables like academic performance and satisfaction in the academic sphere [48,49]. In this line, this investigation was in accordance with those studies, since it included in its results the influence of academic expectations in positive school climate development. On the other hand, regarding future expectations in the academic context, Castrillón-Gómez et al. [50] as well as Hernández-Jácquez and Montes-Ramos [51] registered a higher premature school dropout among demotivated pupils, which might be caused by boring teaching methodologies, inability to pique students’ interest, and also, inability to foster feeling of competence or their need of a relationship with their peers, since students assume a passive role during lessons, limited to memorizing what it is taught by repeating what teachers say or do. This fact highlights, as mentioned before, the importance of teaching methodologies in educational development and the imperative need of a real methodological innovation when objectives are not achieved.

Moreover, with regard to personal responsibility, one of its analyzed components is the ability to set future expectations and create personal plans with intermediate steps to achieve them [47]. A priori, it might seem coherent that a lower capability in this sense could foster premature studies dropout out looking for an early job insertion, which tends to be produced in low social status jobs, with lower remuneration, and analogously, based on physical effort [52]. In this decision of dropping out of ones studies, Rodríguez-Pineda and Andrey [53] remarked that the influence of academic performance, according to the results of the study carried out by Kusnierz et al. [54], is lower among boys than among girls, which is a fact that could explain the higher predilection found in girls toward continuing their studies, even in the university context.

Additionally, regarding the influences of the stage reached in education on future academic expectations, the greater leaning of secondary students toward finishing their studies and starting work might be due to the fact that they are at a moment when that possibility can become a fact, and even more, when their former classmates might have already taken that choice. The different situation among those who could be considered as their equals, who start to earn incomes and are able to afford their purchases, might be attractive to those students who are still studying, promoting studies dropout [55]. This, at the same time, could be exacerbated, according to the conclusions reached by Tomé et al. [56], through the displacement of family influence to a secondary role, due to the growth of the influence of the student’s friends.

Finally, there are certain limitations that should be considered when interpreting these results. First, the data collection instrument, in this case, questionnaires, which, despite the fact that anonymity was ensured, are still subject to the influence of social desirability [57]. The sample was not representative, individuals were not randomly selected to represent an entire group as a whole. Moreover, the participant sample share was of a similar geographical context, with a cultural state that also tended to be similar. Consequently, in future research, we propose the analysis of the influence that parents’ level of studies, students’ socio-economic context, and the prevailing teaching methodologies, have on future academic expectations. In addition, as a criterion of rigor for this future research, we recommend the combination of quantitative (like questionnaires) and qualitative data collection techniques (like interviews or focus groups) with students, parents, and even their teachers.

## 5. Conclusions

The students who have higher future expectances (do a degree or upper vocational formation) have a more autonomous motivation (intrinsic and identified regulation), a higher satisfaction of their basic psychological needs, social and personal responsibility, and the school and teaching climate are also better. On the other hand, lower future expectances (compulsory secondary studies or basic vocational training) suppose higher values in amotivation. Taking into account gender and the educational stage reached, boys and secondary school students have lower future expectations than girls and elementary students. Finally, the satisfaction of basic psychological needs, autonomous motivation, personal and social responsibility, and the school social climate are related to higher academic expectations, with the improvement of these variables being especially important for male and secondary school students who could have a higher possibility of dropping out of school.

## Figures and Tables

**Table 1 ijerph-18-04558-t001:** Descriptive analysis and correlations.

	*M*	*DT*	*A*	*K*	*R*		2	3	4	5	6	7	8	9	10	11	12	13	14
1. Intrinsic Motivation	5.06	1.17	−0.554	−0.179	1–7	0.905	0.566 **	0.710 **	0.405 **	−0.160 **	0.597 **	0.585 **	0.470 **	0.528 **	0.545 **	0.464 **	0.520 **	0.750 **	0.661 **
2. Identified Regulation	5.61	1.06	−0.676	−0.143	1–7	0.644		0.503 **	0.592 **	−0.243 **	0.340 **	0.448 **	0.349 **	0.329 **	0.334 **	0.352 **	0.349 **	0.528 **	0.455 **
3. Introjected Regulation	5.38	1.22	−0.683	−0.169	1–7	0.735			0.460 **	−0.067 *	0.475 **	0.497 **	0.367 **	0.380 **	0.383 **	0.337 **	0.445 **	0.545 **	0.535 **
4. External Regulation	5.90	1.08	−1.090	0.814	1–7	0.732				−0.107 **	0.256 **	0.311 **	0.267 **	0.172 **	0.151 **	0.248 **	0.249 **	0.181**	0.334 **
5. Amotivation	1.94	1.28	1.571	2.060	1–7	0.786					0.028	−0.177 **	−0.153 **	−0.184 **	−0.151 **	−0.201 **	−0.230 **	−0.737 **	−0.121 **
6. Autonomy	3.50	0.81	−0.207	−0.290	1–5	0.565						0.576 **	0.441 **	0.495 **	0.515 **	0.361 **	0.351 **	0.374 **	0.806 **
7. Competence	4.01	0.79	−0.228	0.076	1–5	0.694							0.589 **	0.542 **	0.541 **	0.446 **	0.477 **	0.499 **	0.865 **
8. Relatedness	4.22	0.85	−0.699	0.321	1–5	0.724								0.463 **	0.535 **	0.437 **	0.394 **	0.397 **	0.822 **
9. School Climate	3.68	0.79	−0.392	−0.417	1–5	0.851									0.919 **	0.500 **	0.495 **	0.475 **	0.601 **
10. Teacher Climate	3.89	0.75	−0.543	−0.423	1–5	0.753										0.534 **	0.493 **	0.470 **	0.638 **
11. Social Responsibility	5.10	0.75	−1.077	0.960	1–6	0.844											0.642 **	0.424 **	0.500 **
12. Personal Responsibility	4.97	0.80	−0.883	0.404	1–6	0.744												0.491 **	0.490 **
13. PMI	3.91	0.68	−0.262	−0.372	1–5	0.775													0.509 **
14. SDI	5.84	4.07	−0.503	−0.419	//														

Note. M = Mean; SD = Standard Deviation; A = Asymmetry; K = Kurtosis; R = Range; α = Cronbach alpha; PMI = Psychological mediator index; and SDI = Self-determination index; ** = *p <* 0.001; * = *p <* 0.05; // = Minimum value in this study = −7.80, maximal value = 15.31.

**Table 2 ijerph-18-04558-t002:** Differences in variables according to academic future expectation.

	Finish SCS or BVF and Start to Work		Do High School or MVF		Do a Degree or SVF				
	*M*	*SD*	*M*	*SD*	*M*	*SD*	*F*	*p*	*η2*
Intrinsic Motivation	4.83_a_	1.26	4.87_a_	1.21	5.25_b_	1.09	14.310	<0.001 **	0.028
Identified Regulation	5.32_a_	1.10	5.51_a_	1.06	5.77_b_	1.03	13.743	<0.001 **	0.027
Introjected Regulation	5.24_a_	1.24	5.35_a_	1.26	5.46_a_	1.20	2.159	0.116	0.004
External Regulation	5.85_a_	1.09	5.90_a_	1.02	5.93_a_	1.11	0.333	0.717	0.001
Amotivation	2.47_a_	1.47	2.05_b_	1.26	1.71_c_	1.17	25.776	<0.001 **	0.050
Autonomy	3.48_a_	0.96	3.44_a_	0.80	3.54_a_	0.76	1.398	0.248	0.003
Competence	3.77_a_	0.84	3.93_a_	0.87	4.14_b_	0.71	16.539	<0.001 **	0.033
Relatedness	4.04_a_	0.93	4.14_a_	0.94	4.33_b_	0.76	9.303	<0.001 **	0.019
School Climate	3.48_a_	0.86	3.61_a_	0.77	3.80_b_	0.77	12.397	<0.001 **	0.025
Teacher Climate	3.68_a_	0.84	3.82_a_	0.77	4.01_b_	0.70	15.007	<0.001 **	0.030
Social Responsibility	4.91_a_	0.88	4.99_a_	0.78	5.23_b_	0.67	16.120	<0.001 **	0.032
Personal Responsibility	4.73_a_	0.84	4.81_a_	0.85	5.15_b_	0.72	28.239	<0.001 **	0.054
PMI	3.77_a_	0.77	3.84_a_	0.73	4.00_b_	0.60	10.453	<0.001 **	0.021
SDI	4.15_a_	4.08	5.17_b_	4.03	6.77_c_	3.84	34.643	<0.001 **	0.066
Wilk’s Lamda: 0.878; F = 5.433; *p =* 0.000									

Note. SCS = Secondary Compulsory Studies; BVF = Basic vocational training; MVF = Middle vocational training; SVF = Superior vocational training; M = Mean; SD = Standard Deviation; F = multivariate value; PMI = Psychological mediator index; and η2 = effect size. ** = *p <* 0.001; a, b, c = group differences using post-hoc test.

**Table 3 ijerph-18-04558-t003:** Differences according to gender and elementary or secondary school.

	Finish SCS or BVF and Start to Work			Do High School or MVF			Do a Degree or SVF					
	Total	%	R	Total	%	R	Total	%	R	x^2^	df	*p*
Men	129	24.3%	6.1	159	29.9%	0.4	243	45.8%	−5.0	42.612	2	<0.001
Women	43	9.5%	−6.1	130	28.7%	−0.4	280	61.8%	5.0			
Elementary school	15	6.1%	−5.4	42	17.1%	−4.9	189	76.8%	8.6	75.798	2	<0.001
Secondary school	157	21.3%	5.4	247	33.5%	4.9	334	45.3%	−8.6			

Note. SCS = Secondary Compulsory Studies; BVF: Basic vocational training; MVF = Middle vocational training; SVF = Superior vocational training; R = Standardized Residual; SD = Standard Deviation; PBN = Psychological Basic Needs; x^2^ = chi squared; df = degree freedom; and *p = p*-value.

## Data Availability

https://osf.io/guf6d/?view_only=d486da8c9ecd417f964896cf43ea423f, accessed on 19 January 2021.

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
