# Peer review of "Future Academic Expectations and Their Relationship with Motivation, Satisfaction of Psychological Needs, Responsibility, and School Social Climate: Gender and Educational Stage"

_ijerph, 2021, doi:10.3390/ijerph18094558_

Round 1
Reviewer 1 Report
the unique part of this study is its focus on future expectations. It shows how high quality motivation and need satisfaction is associated with higher academic goals, whereas amotivation and low need satisfaction is oppositely related
a second important finding concerns the association between males versus females and lower expectations. It would be good to directly report a mediation analysis to show this can be accounted for by motivation variables, which is an implied finding
the authors do a good job of describing the strengths and limitations, the latter having to do with cross-sectional, self-report methods.
small note, but this doesn't seem to be an application of Vallerand's (1997) hierarchical model as described, but instead is focused on variable relationships within a single domain. and level of analysis.
Author Response
Dear Editor and reviewer:
First of all, we would like to thanks the reviewer contributions to increase the quality of the manuscript.
Comments and Suggestions for Authors
Point 1: The unique part of this study is its focus on future expectations. It shows how high quality motivation and need satisfaction is associated with higher academic goals, whereas amotivation and low need satisfaction is oppositely related.
Response 1: Thank you for your appreciation.
Point 2: A second important finding concerns the association between males versus females and lower expectations. It would be good to directly report a mediation analysis to show this can be accounted for by motivation variables, which is an implied finding.
Response 2: This is a really interesting issue that we have not covered in this study, but we will take into account for future ones, because believe this procedure would involve a new objective and a new manuscript framework should be implemented. Furthermore, there are some issues we have to learn more thoughtfully. We confess we have done a preliminary mediation analysis, with macro PROCESS for SPSS (Hayes, 2013), having autonomous motivation as independent variable, future expectations as a dependent variable and gender as a mediator. However, the outcomes inform we cannot run this with a dichotomous mediator. We keep working in this kind of analyses to learn more about this idea. Thank you very much.
Point 3: The authors do a good job of describing the strengths and limitations, the latter having to do with cross-sectional, self-report methods.
Response 3: Thank you.
Point 4: Small note, but this doesn't seem to be an application of Vallerand's (1997) hierarchical model as described, but instead is focused on variable relationships within a single domain. and level of analysis.
Response 4: We wanted to give a theoretical framework for our research by pointed out the Self-Determination Theory (SDT) and justifying why we have chosen these variables. In this sense, the horizontal organization of the Vallerand’s hierarchical model reflects the social psychological processes. It suggests a causal sequence of events involving social factors, psychological needs, motivation, and outcomes (Vallerand & Lalande, 2011). We believe these ideas help readers to understand the background.
Vallerand, R. J., & Lalande, D. R. (2011). The MPIC Model: the perspective of the hierarchical model of intrinsic and extrinsic motivation. Psychological Inquiry, 22, 45-51. Doi: 10.1080/1047840X.2011.545366
Reviewer 2 Report
- I would appreciate if authors review the last Educational Law in Spain, as the one referenced by them is already obsolete (reference [1])
- I find difficult to accomplish the part of the secondary objective about the differences in future expectations according to the educational stage, because the sample is not significative in this sense as there are three times more secondary students (738) than primary ones (246) although, in Spain, statistically there are more primary students than secundary ones (six courses versus four courses)
- I find neccessary to re-write the procedure in order to make it clearer. In this sense it needs to follow a chronological procedure, with a major explanation of some of the stepts, as how is it the presentation the students had to watch?, how were the questionnaries implemented? Were they individually implemented? Classroom implemented? school implemented? all of participants at the same time? In which period were the instruments implemented?
- About statistical analysis, I am not sure if skewness and kurtosis are enough to ensure normality of data, why did not the authors use the Kolmogorov-Smirnoff analysis with Lilliefors corrections to ensure normality of data? I am almost sure that if authors use this type of analysis the data would not follow a normal distribution, so analysis they should use must be non parametrical.
Author Response
Dear Editor and reviewer:
First of all, we would like to thanks the reviewer contributions to increase the quality of the manuscript.
Comments and Suggestions for Authors
Point 1: I would appreciate if authors review the last Educational Law in Spain, as the one referenced by them is already obsolete (reference [1]).
Response 1: The new education law had been included. Line 353.
Point 2: I find difficult to accomplish the part of the secondary objective about the differences in future expectations according to the educational stage, because the sample is not significative in this sense as there are three times more secondary students (738) than primary ones (246) although, in Spain, statistically there are more primary students than secundary ones (six courses versus four courses).
Response 2: This is a limitation of our study. In “Study design and participants”, inside the “Materials and Methods” section, it is stated that sample was chosen by accessibility and convenience. Also, we have added a new sentence in the last part of the manuscript “Limitations” noting that the sample was not representative and for future studies should be randomly selected, warning readers about this issue. Lines 316-317.
Point 3: I find neccessary to re-write the procedure in order to make it clearer. In this sense it needs to follow a chronological procedure, with a major explanation of some of the stepts, as how is it the presentation the students had to watch?, how were the questionnaries implemented? Were they individually implemented? Classroom implemented? school implemented? all of participants at the same time? In which period were the instruments implemented?
Response 3: The procedure had been clarified following the reviewer suggestions adding this paragraph. Lines 127-136: “Subsequently, they were informed of the purpose of the research and were told that participation was voluntary and confidential. Students attended an oral presentation about how to complete the questionnaires with an explanation of that by one of the researchers. After that, the teacher read the items of each scale in order to ensure their understanding. The questionnaires were administered in the specific classroom of each grade within their school, in a calm environment and without the presence of possible distractions. The teacher and one of the researchers stayed with them all the time solving possible doubt. The participants were requested to provide true answers. The test was completed by each class-group in approximately 30 minutes all students at the same time”.
Point 4: About statistical analysis, I am not sure if skewness and kurtosis are enough to ensure normality of data, why did not the authors use the Kolmogorov-Smirnoff analysis with Lilliefors corrections to ensure normality of data? I am almost sure that if authors use this type of analysis the data would not follow a normal distribution, so analysis they should use must be non parametrical.
Response 4: The reviewer is right. We checked the normal distribution of the data and the results were not all the parametric statistical assumptions were confirmed. Likewise, we performed the analyses of Kruskal-Wallis as well, obtaining very similar result with both procedures.
Three new sentences have been added clarifying this issue. Lines 193-198.